# Stochastic Weight Averaging in Parallel: Large-Batch Training That Generalizes Well

**Vipul Gupta**[*][†]
vipul_gupta@berkeley.edu
Deparment of EECS, UC Berkeley

**Santiago Akle Serrano** [*]
sakle@apple.com
Apple Inc.

**Dennis DeCoste**
ddecoste@apple.com
Apple Inc.

## Abstract

We propose Stochastic Weight Averaging in Parallel (SWAP), an algorithm to accelerate DNN training. Our algorithm uses large mini-batches to compute an approximate solution quickly and then refines it by averaging the weights of multiple models computed independently and in parallel. The resulting models generalize equally well as those trained with small mini-batches but are produced in a substantially shorter time. We demonstrate the reduction in training time and the good generalization performance of the resulting models on the computer vision datasets CIFAR10, CIFAR100, and ImageNet.

## 1 Introduction

Stochastic gradient descent (SGD) and its variants are the de-facto methods to train deep neural networks (DNNs). Each iteration of SGD computes an estimate of the objective's gradient by sampling a *mini-batch* of the available training data and computing the gradient of the loss restricted to the sampled data. A popular strategy to accelerate DNN training is to increase the mini-batch size together with the available computational resources. Larger mini-batches produce more precise gradient estimates; these allow for higher learning rates and achieve larger reductions of the training loss per iteration. In a distributed setting, multiple nodes can compute gradient estimates simultaneously on disjoint subsets of the mini-batch and produce a consensus estimate by averaging all estimates, with one synchronization event per iteration. Training with larger mini-batches requires fewer updates, thus fewer synchronization events, yielding good overall scaling behavior.

Even though the training loss can be reduced more efficiently, there is a maximum batch size after which the resulting model tends to have worse generalization performance (McCandlish et al., 2018; Keskar et al., 2016; Hoffer et al., 2017; Golmant et al., 2018; Shallue et al., 2018). This phenomenon forces practitioners to use batch sizes below those that achieve the maximum throughput and limits the usefulness of large-batch training strategies.

Stochastic Weight Averaging (SWA) (Izmailov et al., 2018) is a method that produces models with good generalization performance by averaging the *weights* of a set of models sampled from the final stages of a training run. As long as the models all lie in a region where the population loss is mostly convex, the average model can behave well, and in practice, it does.

We have observed that if instead of sampling multiple models from a sequence generated by SGD, we generate multiple independent SGD sequences and average models from each, the resulting model achieves similar generalization performance. Furthermore, if all the independent sequences use small-batches, but start from a model trained with large-batches, the resulting model achieves generalization performance comparable with a model trained solely with small-batches. Using these observations, we derive *Stochastic Weight Averaging in Parallel* (SWAP): A simple strategy to accelerate DNN training by better utilizing available compute resources. Our algorithm is simple to implement, fast and produces good results with minor tuning.

For several image classification tasks on popular computer vision datasets (CIFAR10, CIFAR100, and ImageNet), we show that SWAP achieves generalization performance comparable to models trained with small-batches but does so in time similar to that of a training run with large-batches. We use SWAP on some of the most efficient publicly available models to date, and show that it's

---

[*]Equal contribution
[†]Work done during an internship at Apple Inc.

able to substantially reduce their training times. Furthermore, we are able to beat the state of the art for CIFAR10 and train in $68\%$ of the time of the winning entry of the DAWNBench competition.[1]

## 2  RELATED WORK

The mechanism by which the training batch size affects the generalization performance is still unknown. A popular explanation is that because of the reduced noise, a model trained using larger mini-batches is more likely to get stuck in a sharper global minima. In (Keskar et al., 2016), the authors argue that sharp minima are sensitive to variations in the data because slight shifts in the location of the minimizer will result in large increases in average loss value. However, if flatness is taken to be the curvature as measured by the second order approximation of the loss, then counterexamples exist. In (Dinh et al., 2017), the authors transform a flat minimizer into a sharp one without changing the behavior of the model, and in (Li et al., 2018), the authors show the reverse behavior when weight-decay is not used.

In (McCandlish et al., 2018), the authors predict that the batch size can be increased up to a *critical size* without any drop in accuracy and empirically validate this claim. For example, the accuracy begins to drop for image classification on CIFAR10 when the batch sizes exceed 1k samples. They postulate that when the batch size is large, the mini-batch gradient is close to the full gradient, and further increasing the batch size will not significantly improve the signal to noise ratio.

In (Hoffer et al., 2017), the authors argue that, for a fixed number of epochs, using a larger batch size implies fewer model updates. They argue that changing the number of updates impacts the distance the weights travel away from their initialization and that this distance determines the generalization performance. They show that by training with large-batches for longer times (thus increasing the number of updates), the generalization performance of the model is recovered. Even though this large-batch strategy generates models that generalize well, it does so in more time than the small-batch alternative.

Irrespective of the generalization performance, the batch size also affects the optimization process. In (Ma et al., 2017), the authors show that for convex functions in the over-parameterized setting, there is a critical batch size below which an iteration with a batch size of $M$ is roughly equivalent to $M$ iterations with a batch size of one, and batch-sizes larger than $M$ do not improve the rate of convergence.

Methods which use adaptive batch sizes exist (Devarakonda et al., 2017; Goyal et al., 2017; Jia et al., 2018; Smith et al., 2017; You et al., 2017). However, most of these methods are either designed for specific datasets or require extensive hyper-parameter tuning. Furthermore, they ineffectively use the computational resources by reducing the batch size during part of the training.

Local SGD (Zhang et al., 2016; Stich, 2018; Li et al., 2019; Yu et al., 2019) is a distributed optimization algorithm that trades off gradient precision with communication costs by allowing workers to independently update their models for a few steps before synchronizing. Post-local SGD (Lin et al., 2018) is a variant, which refines the output of large-batch training with local-SGD. The authors have observed that the resulting model has better generalization than the model trained with large-batches and that their scheme achieves significant speedups. In this manner Post-local SGD is of a very similar vein than the present work. However, while Post-local SGD lets the models diverge for $T$ iterations where $T$ is in the order of tens, SWAP averages the models once after multiple epochs. For example, in our Imagenet exeperiments (see Sec. 5) we average our models after tens of thousands of updates, while Post-local SGD does after at most 32. Because of this difference, we believe that the mechanisms that power the success of SWAP and Post-local SGD must be different and point to different phenomena in DNN optimization.

Stochastic weight averaging (SWA) (Izmailov et al., 2018) is a method where models are sampled from the later stages of an SGD training run. When the weights of these models are averaged, they result in a model with much better generalization properties. This strategy is very effective and has been adopted in multiple domains: deep reinforcement learning (Nikishin et al.), semi-supervised learning (Athiwaratkun et al., 2019), Bayesian inference (Maddox et al., 2019), low-precision training (Yang et al., 2019). In this work, we adapt SWA to accelerate DNN training.

---

[1]The https://dawn.cs.stanford.edu/benchmark/

## 3 STOCHASTIC WEIGHT AVERAGING IN PARALLEL

We describe SWAP as an algorithm in three phases (see Algorithm 1): In the first phase, all workers train a single model by computing large mini-batch updates. Synchronization between workers is required at each iteration and a higher learning rate is used. In the second phase, each worker independently refines its copy of the model to produce a different set of weights. Workers use a smaller batch size, a lower learning rate, and different randomizations of the data. No synchronization between workers is required in this phase. The last phase consists of averaging the weights of the resulting models and computing new batch-normalization statistics to produce the final output.

Phase 1 is terminated before the training loss reaches zero or the training accuracy reaches 100% (for example, a few percentage points below 100%). We believe that stopping early precludes the optimization from getting stuck at a location where the gradients are too small and allows the following stage to improve the generalization performance. However, the optimal stopping accuracy is a hyper-parameter that requires tuning.

During phase 2, the batch size is appropriately reduced and small-batch training is performed independently and simultaneously. Here, each worker (or a subset of them) performs training using all the data, but sampling in different random order. Thus, after the end of the training process, each worker (or subset) will have produced a different model.

Figure 1 plots the accuracies and learning-rate schedules for a run of SWAP. During the large-batch phase (phase 1), all workers share a common model and have the same generalization performance. During the small-batch phase (phase 2) the learning rates for all the workers are the same but their testing accuracies differ as the stochasticity causes the models to diverge from each other. We also plot the test-accuracy of the averaged model that would result were we to stop phase 2 at that point. Note that the averaged model performs consistently better than *each individual model*.

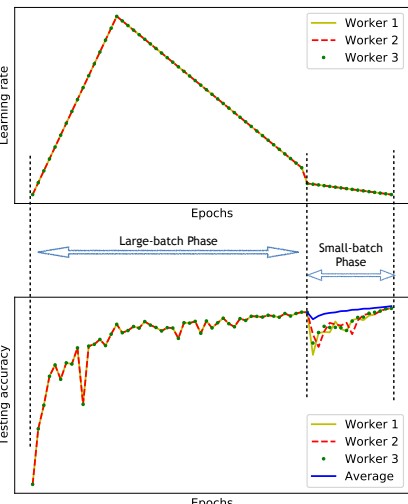

Figure 1: Learning rate schedules and CIFAR10 test accuracies for workers participating in SWAP. The large-batch phase with synchronized models is followed by the small-batch phase with diverging independent models. The test accuracy of the averaged weight model is computed by averaging the independent models and computing the test loss for the resulting model.

## 4 LOSS LANDSCAPE VISUALIZATION AROUND SWAP ITERATES

To visualize the mechanism behind SWAP, we plot the error achieved by our test network on a plane that contains the outputs of the three different phases of the algorithm. Inspired by (Garipov et al., 2018) and (Izmailov et al., 2018), we pick orthogonal vectors $u, v$ that span the plane which contains $\theta_1, \theta_2, \theta_3$. We plot the loss value generated by model $\theta = \theta_1 + \alpha u + \beta v$ at the location $(\alpha, \beta)$. To plot a loss value, we first generate a weight vector $\theta$, compute the batch-norm statistics for that model (through one pass over the training data), and then evaluate the test and train accuracies.

In Figure 2, we plot the training and testing error for the CIFAR10 dataset. Here 'LB' marks the output of phase one, 'SGD' the output of a single worker after phase two, and 'SWAP' the final

---

**Algorithm 1:** Stochastic Weight Averaging in Parallel (SWAP)

---

1   Number of workers $W$; Weight initialization $\theta_0$; $t = 0$
2   Training accuracy, $\tau$, at which to exit phase one
3   Learning rate schedules $LR_1$ and $LR_2$ for phase one and two, respectively
4   Mini-batch sizes $B_1$ and $B_2$ for phase one and two, respectively
5   Gradient of loss function for sample $i$ at weight $\theta$: $g^i$
6   SGDUpdate($\cdot$) : A function that updates the weights using SGD with momentum and weight decay
7   **Phase 1:**
8   **while** *Training accuracy* $\leq \tau$ **do**
9      $\eta \leftarrow LR_1(t)$
10      **for** $w$ *in* $[0, ..., W-1]$ *In parallel* **do**
11          $B^w \leftarrow$ random sub-sample of training data with size $\frac{B_1}{W}$
12          $g^w \leftarrow \frac{W}{|B_1|} \sum_{i \in B^w} g^i$ worker gradient
13      **end**
14      $g_t \leftarrow \frac{1}{W} \sum g^w$ synchronization of worker gradients
15      $\theta_{t+1} \leftarrow \theta_t + \text{SGDUpdate}(\eta_t, g_t, g_{t-1}, \cdots)$;    /* first order method update */
16      $t = t + 1$; $T = t$
17   **end**
18   **Phase 2:**
19   **for** $t$ *in* $[T, T+Q]$ **do**
20      $\eta \leftarrow LR_2(t - T)$
21      **for** $w$ *in* $[0, ..., W-1]$ *In parallel* **do**
22          $B^w \leftarrow$ random sub-sample of training data with size $B_2$
23          $g^w \leftarrow \frac{1}{|B_2|} \sum_{i \in B^w} g^i$ worker gradient
24          $\theta_{t+1}^w \leftarrow \theta_t^w + \text{SGDUpdate}(\eta_t, g_t^w, g_{t-1}^w, \cdots)$;    /* first order method update at local worker */
25      **end**
26   **end**
    /* We get $W$ different models at the end of phase 2        */
27   **Phase 3:** $\hat{\theta}_\ell \leftarrow \frac{1}{W} \sum \theta_{T+Q}^i$ produce averaged model
28   Compute batch-norm statistics for $\hat{\theta}_\ell$ to produce $\theta_\ell$
    **Result:** Final model $\theta_\ell$

---

model. Color codes correspond to error measures at the points interpolated on the plane. In Figure 2a, we observe that the level-sets of the training error (restricted to this plane) form an almost convex basin and that both the output of phase 1 ('LB')[2] and the output of one of the workers of phase 2 ('SGD') lie in the outer edges of the basin. Importantly, during phase 2 the model traversed to a *different side* of the basin (and not to the center). Also, the final model ('SWAP') is closer to the center of the basin.

When we visualize these three points on the test loss landscape (Figure 2b), we observe that the variations in the topology of the basin cause the 'LB' and 'SGD' points to fall in regions of higher error. But, since the 'SWAP' point is closer to the center of the basin, it is less affected by the change in topology. In Figure 3, we neglect the 'LB' point and plot the plane spanned by three workers 'SGD1', 'SGD2', 'SGD3'. In Figure 3a, we can observe that these points lie at different sides of the training error basin while 'SWAP' is closer to the center. In Figure 3b, we observe that the change in topology causes the worker points to lie in regions of higher testing errors than 'SWAP', which is again close to the center of both basins. For reference, we have also plotted the best model that can be generated by this region of the plane.

---

[2]Recall that the weights 'LB' are obtained by stopping the large-batch training early in phase 1. Hence, the training error for 'LB' is worse than 'SGD' and 'SWAP'.

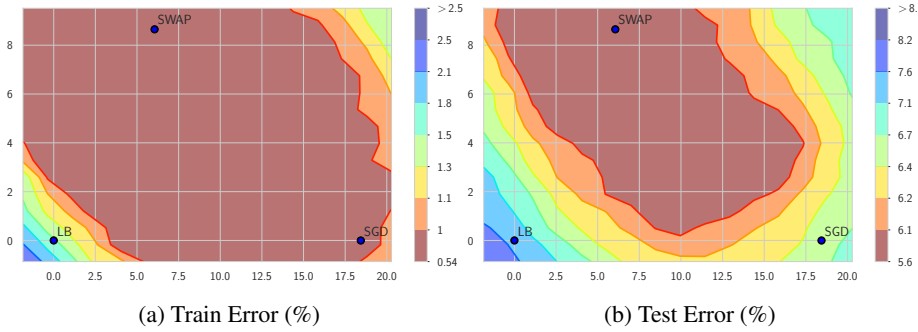

(a) Train Error (%)                    (b) Test Error (%)

Figure 2: CIFAR10 train and test error restricted to a 2D plane spanned by the output of phase 1 ('LB'), one of the outputs of phase 2 ('SGD') and the averaged model ('SWAP').

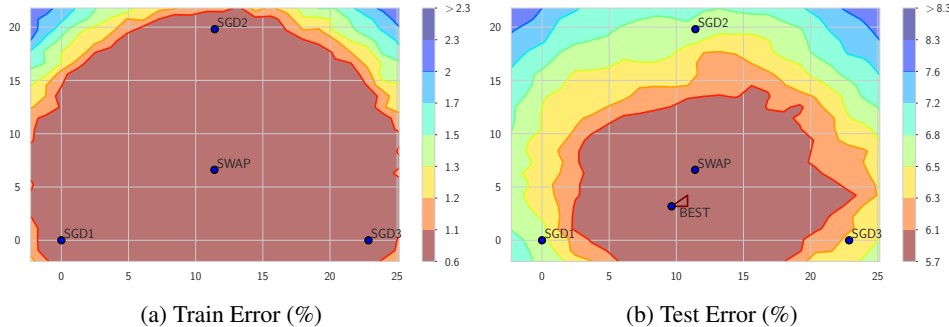

(a) Train Error (%)                    (b) Test Error (%)

Figure 3: CIFAR10 train and test error restricted to a 2D plane spanned by the output of three workers after phase 2 ('SGD1', 'SGD2', 'SGD3') and location of the average model ('SWAP'). The minimum test error achievable for models restricted to this region of the plane (marked as *BEST*).

### 4.1 SAMPLING FROM INDEPENDENT RUNS OF SGD OR SAMPLING FROM ONE

In (Mandt et al., 2017), the authors argue that in the later stages of SGD the weight iterates behave similar to an Ornstein Uhlenbeck process. So, by maintaining a constant learning rate the SGD iterates should reach a stationary distribution that is similar to a high-dimensional Gaussian. This distribution is centered at the local minimum, has a covariance that grows proportionally with the learning rate, inversely proportional to the batch size and has a shape that depends on both the Hessian of the mean loss and covariance of the gradient.

The authors of (Izmailov et al., 2018) argue that by virtue of being a high dimensional Gaussian all the mass of the distribution is concentrated near the 'shell' of the ellipsoid, and therefore, it is unlikely for SGD to access the interior. They further argue that by sampling weights from an SGD run (leaving enough time steps between them) will choose weights that are spread out on the surface of this ellipsoid and their average will be closer to the center.

Without any further assumptions, we can justify sampling from different SGD runs (as done in phase 2 during SWAP). As long as all runs start in the same basin of attraction, and provided the model from (Mandt et al., 2017) holds, all runs will converge to the same stationary distribution, and each run can generate independent samples from it.

### 4.2 ORTHOGONALITY OF THE GRADIENT AND THE DIRECTION TO THE CENTER OF BASIN

To win some intuition on the advantage that SWA and SWAP have over SGD, we measure the cosine similarity between the gradient descent direction, $-g_i$, and the direction towards the output of SWAP, $\Delta\theta = \theta_{\text{swap}} - \theta_i$. In Figure 4, we see that the cosine similarity, $\frac{\langle \Delta\theta, -g_i \rangle}{\|g_i\|\|\Delta\theta\|}$, decreases as the training enters its later stages. We believe that towards the end of training, the angle between the gradient direction and the directions toward the center of the basin is large, therefore the process moves mostly orthogonally to the basin, and progress slows. However, averaging samples from different sides of the basin can (and does) make faster progress towards the center.

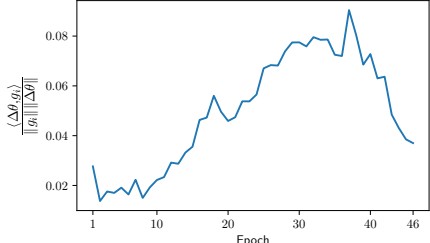

Figure 4: Cosine similarity between direction of gradient descent and $\Delta\theta$

# 5 EXPERIMENTS

In this section we evaluate the performance of SWAP for image classification tasks on the CIFAR10, CIFAR100, and ImageNet datasets.

## 5.1 CIFAR10 AND CIFAR100

For the experiments in this subsection, we found the best hyper-parameters using grid searches (see Appendix A for details). We train using mini-batch SGD with Nesterov momentum (set to 0.9) and weight decay of $5\times10^{-4}$. We augment the data using cutout (DeVries & Taylor, 2017) and use a fast-to-train custom ResNet 9 from a submission [3] to the DAWNBench leaderboard (Coleman et al.). All experiments were run on one machine with 8 NVIDIA Tesla V100 GPUs and use Horovod (Sergeev & Del Balso, 2018) to distribute the computation. All statistics were collected over 10 different runs.

**CIFAR10**: For these experiments, we used the following settings—SWAP phase one: 4096 samples per batch using 8 GPUs (512 samples per GPU). Phase one is terminated when the training accuracy reaches 98% (on average 108 epochs). SWAP phase two: 8 workers with one GPU each and 512 samples per batch for 30 epochs. The experiment that uses only large-batches had 4096 samples per batch across 8 GPUs and is run for 150 epochs. The experiments that use only small-batches had 512 samples per batch on 2 GPUs and is trained for 100 epochs.

Table 1 compares the best test accuracies and corresponding training times for models trained with small-batch only, with large-batch only, and with SWAP. We report the average accuracy of the workers before averaging and the accuracy of the final model.

| CIFAR10 | Test Accuracy (%) | Training Time (sec) |
|---|---|---|
| SGD (small-batch) | $95.24 \pm 0.09$ | $254.12 \pm 0.62$ |
| SGD (large-batch) | $94.77 \pm 0.23$ | $132.62 \pm 1.09$ |
| SWAP (before averaging) | $94.70 \pm 0.20$ | $167.57 \pm 3.25$ |
| SWAP (after averaging) | $95.23 \pm 0.08$ | $169.20 \pm 3.25$ |

Table 1: Training Statistics for CIFAR10

**CIFAR100**: For these experiments, we use the following settings—SWAP phase one: 2048 samples per batch using 8 GPUs (256 samples per GPU). Phase one exits when the training accuracy reaches 90% (on average 112 epochs). SWAP phase two: 8 workers with one GPU each and 128 samples per batch, training for for 10 epochs. The experiments that use only large-batch training were run for 150 epochs with batches of 2048 on 8 GPUs The experiments that use only small-batch were trained for 150 epochs using batches of 128 on 1 GPU.

| CIFAR100 | Test Accuracy (%) | Training Time (sec) |
|---|---|---|
| SGD (small-batch) | $77.01 \pm 0.25$ | $573.76 \pm 2.25$ |
| SGD (large-batch) | $75.84 \pm 0.35$ | $116.13 \pm 1.35$ |
| SWAP (before averaging) | $75.74 \pm 0.15$ | $123.11 \pm 1.85$ |
| SWAP (after averaging) | $78.18 \pm 0.21$ | $125.34 \pm 1.85$ |

Table 2: Training Statistics for CIFAR100

Table 2 compares the best test accuracies and corresponding training times for models trained with only small-batches (for 150 epochs), with only large-batches (for 150 epochs), and with SWAP.

---

[3]https://github.com/davidcpage/cifar10-fast

For SWAP, we report test accuracies obtained using the last SGD iterate before averaging, and test accuracy of the final model obtained after averaging. We observe significant improvement in test accuracies after averaging the models.

For both CIFAR 10 and CIFAR100, training with small-batches achieves higher testing accuracy than training with large-batches but takes much longer to train. SWAP, however, terminates in time comparable to the large-batch run but achieves accuracies on par (or better) than small batch training.

**Achieving state of the art training speeds for CIFAR10**: At the time of writing the front-runner of the DAWNBench competition takes 37 seconds with 4 Tesla V100 GPUs to train CIFAR10 to 94% test accuracy. Using SWAP with 8 Tesla V100 GPUs, a phase one batch size of 2048 samples and 28 epochs, and a phase two batch size of 256 samples for one epoch is able to reach the same accuracy in 27 seconds.

## 5.2 EXPERIMENTS ON IMAGENET

We use SWAP to accelerate a publicly available fast-to-train ImageNet model with published learning rate and batch size schedules [4]. The default settings for this code modify the learning-rates and batch sizes throughout the optimization (see Figure 5). Our small-batch experiments train ImageNet for 28 epochs using the published schedules with no modification and are run on 8 Tesla V100 GPUs. Our large-batch experiments modify the schedules by doubling the batch size and doubling the learning rates (see Figure 5) and are run on 16 Tesla V100 GPUs. For SWAP phase 1, we use the large-batch settings for 22 epochs, and for SWAP phase 2, we run two independent workers each with 8 GPUs using the settings for small-batches for 6 epochs.

We observe that doubling the batch size reduces the Top1 and Top5 test accuracies with respect to the small-batch run. SWAP, however, recovers the generalization performance at substantially reduced training times. Our results are compiled in Table 3 (the statistics were collected over 3 runs). We believe it's worthy of mention that these accelerations were achieved with no tuning other than increasing the learning rates proportionally to the increase in batch size and reverting to the original schedule when transitioning between phases.

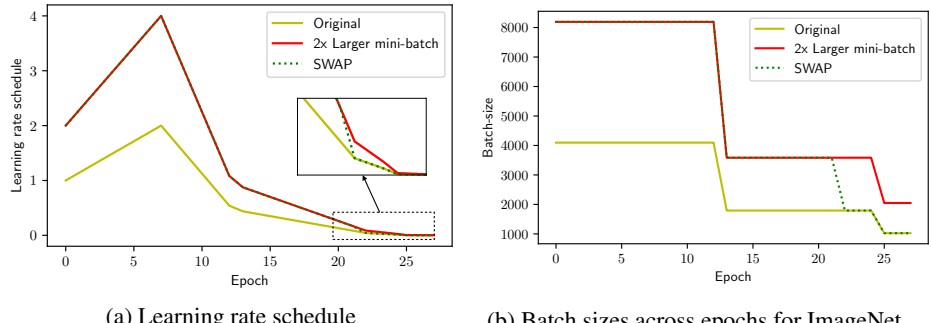

(a) Learning rate schedule          (b) Batch sizes across epochs for ImageNet

Figure 5: Learning rate and mini-batch schedules used for ImageNet. The original schedule for 8 GPUs was taken from an existing DAWNBench submission. For a larger batch experiment, we double the batch size, double the number of GPUs and double the learning rate of the original schedule. For SWAP, we switch from the modified schedule to the original schedule as we move from phase 1 to phase 2.

| ImageNet | Top1 Accuracy (%) | Top5 Accuracy (%) | Training Time (min) |
|---|---|---|---|
| SGD (small-batch) | $76.14 \pm 0.07$ | $93.30 \pm 0.07$ | $235.29 \pm 0.33$ |
| SGD (large-batch) | $75.86 \pm 0.03$ | $92.98 \pm 0.06$ | $127.20 \pm 0.78$ |
| SWAP (before averaging) | $75.96 \pm 0.02$ | $93.15 \pm 0.02$ | $149.12 \pm 0.55$ |
| SWAP (after averaging) | $76.19 \pm 0.03$ | $93.32 \pm 0.02$ | $156.55 \pm 0.56$ |

Table 3: Training Statistics for ImageNet

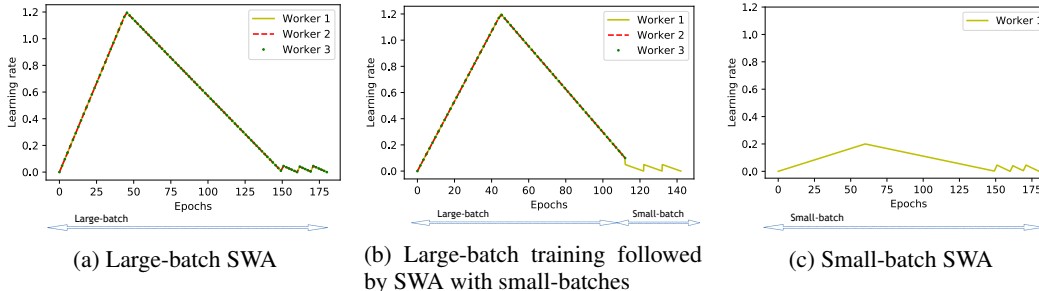

(a) Large-batch SWA     (b) Large-batch training followed by SWA with small-batches     (c) Small-batch SWA

Figure 6: Illustration of SWA with different batch sizes

## 5.3 EMPIRICAL COMPARISON OF SWA AND SWAP

We now compare SWAP with SWA: the sequential weight averaging algorithm from Izmailov et al. (2018). For the experiments in this section, we use the CIFAR100 dataset. We sample the same number of models for both SWA and SWAP and maintain the same number of epochs per sample. For SWA, we sample each model with 10 epochs in-between and average them to get the final model. For SWAP, we run 8 independent workers for 10 epochs each and use their average as the final model.

**Large-batch SWA**: We explore if SWA can recover the test accuracy of small-batch training on a large-batch training run. We use the same (large) batch size throughout. We follow an initial training cycle with cyclic learning rates (with cycles of 10 epochs) to sample 8 models (one from the end of each cycle). See Figure 6a for an illustration of the learning rate schedule.

As expected we observe that the large-batch training run achieves lower training accuracy, but surprisingly SWA was unable to improve it (see Table 4, row 1).

**Large-batch followed by small-batch SWA**: We evaluate the effect of executing SWA using small-batches after a large-batch training run. We interrupt the large-batch phase at the same accuracy we interrupt phase 1 of our CIFAR100 experiment (Table 2). In this case, the small-batch phase uses a single worker and samples the models sequentially. SWA is able to reach the test accuracy of a small-batch run but requires more than three times longer than SWAP to compute the model (see Table 4, row 2). An illustration of the learning rate schedule is provided in Figure 6b.

**Small-batch SWA and SWAP**: We start the SWA cyclic learning rate schedule from the best model found by solely small-batch training (table 2, row 1). Since the cycle length and cycle count are fixed, the only free parameter is the peak learning rate. We select this using a grid-search. Once the SWA schedule is specified, we re-use the peak learning rate settings in SWAP. We start phase two from the model that was generated as the output of phase 1 for the experiment on section 5.1 reported on table 2 rows 3 and 4. With these settings, small-batch SWA achieves better accuracy than SWAP (by around $\sim 0.9\%$) at 6.8x more training time.

Next, we wish to explore the speed-up that SWAP achieves over SWA if the precision of SWA is set as a target. To that end, we relax the constraints on SWAP. By increasing the phase two schedule from one 10 epoch cycle to two 20 epoch cycles and sampling two models from each worker (16 models) the resulting model achieved a test accuracy of $79.11\%$ in 241 seconds or 3.5x less time.

| CIFAR100 | Test accuracy before averaging (%) | Test accuracy after averaging (%) | Training Time (sec) |
|---|---|---|---|
| Large-batch SWA | $76.06 \pm 0.25$ | $76.00 \pm 0.31$ | $376.4 \pm 2.25$ |
| Large-batch followed by small-batch SWA | $76.26 \pm 0.35$ | $78.12 \pm 0.14$ | $398.0 \pm 1.35$ |
| Small-batch SWA | $76.80 \pm 0.15$ | $79.09 \pm 0.19$ | $848.6 \pm 5.61$ |
| SWAP (10 small-batch epochs) | $75.74 \pm 0.15$ | $78.18 \pm 0.21$ | $125.30 \pm 1.85$ |
| SWAP (40 small-batch epochs) | $76.19 \pm 0.19$ | $79.11 \pm 0.12$ | $241.54 \pm 1.62$ |

Table 4: Comparison: SWA versus SWAP

---

[4]Available at https://github.com/cybertronai/imagenet18_old

## 6    CONCLUSIONS AND FUTURE WORK

We propose Stochastic Weight Averaging in Parallel (SWAP), an algorithm that uses a variant of Stochastic Weight Averaging (SWA) to improve the generalization performance of a model trained with large mini-batches. Our algorithm uses large mini-batches to compute an approximate solution quickly and then refines it by averaging the weights of multiple models trained using small-batches. The final model obtained after averaging has good generalization performance and is trained in a shorter time. We believe that this variant and this application of SWA are novel.

We observed that using large-batches in the initial stages of training does not preclude the models from achieving good generalization performance. That is, by refining the output of a large-batch run, with models sampled sequentially as in SWA or in parallel as in SWAP, the resulting model is able to perform as well as the models trained using small-batches only. We confirm this in the image classification datasets CIFAR10, CIFAR100, and ImageNet.

Through visualizations, we complement the existing evidence that averaged weights are closer to the center of a training loss basin than the models produced by stochastic gradient descent. It's interesting to note that the basin into which the large mini-batch run is converging to seems to be the same basin where the refined models are found. So, it is possible that regions with bad and good generalization performance are connected through regions of low training loss and, more so, that both belong to an almost convex basin. Our method requires the choice of (at least) one more hyperparameter: the transition point between the large-batch and small-batch. For our experiments, we chose this by using a grid search. A principled method to choose the transition point will be the focus of future work.

In future work we intend to explore the behavior of SWAP when used with other optimization schemes, such as Layer-wise Adaptive Rate Scaling (LARS) (You et al., 2017), mixed-precision training Jia et al. (2018), post-local SGD (Lin et al., 2018) or NovoGrad (Ginsburg et al., 2019). The design of SWAP allows us to substitute any of these for the large-batch stage, for example, we can use local SGD to accelerate the first stage of SWAP by reducing the communication overhead.

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

## A   HYPERPARAMETERS FOR CIFAR10 AND CIFAR100 EXPERIMENTS

We provide the parameters used in the experiments of Section 5.1. These were obtained by doing independent grid searches for each experiment. For all CIFAR experiments, the momentum and weight decay constants were kept at 0.9 and $5 \times 10^{-4}$ respectively. Tables 5 and 6 list the remaining hyperparameters. When a stopping accuracy of $100\%$ is listed, we mean that the maximum number of epochs were used.

| CIFAR10 | SGD (small-batch) | SGD (large-batch) | SWAP (Phase 1) | SWAP (Phase 2) |
|---|---|---|---|---|
| Batch-size | 512 | 4096 | 4096 | 512 |
| Learning-rate Peak | 0.3 | 1.2 | 1.2 | 0.12 |
| Maximum Epochs | 100 | 150 | 150 | 30 |
| Warm-up Epochs | 30 | 30 | 30 | 0 |
| GPUs used per model | 2 | 8 | 8 | 1 |
| Stopping Accuracy (%) | 100 | 100 | 98 | 100 |

Table 5: Hyperparameters obtained using tuning for CIFAR10

| CIFAR100 | SGD (small-batch) | SGD (large-batch) | SWAP (Phase 1) | SWAP (Phase 2) |
|---|---|---|---|---|
| Batch-size | 128 | 2048 | 2048 | 128 |
| Learning-rate Peak | 0.2 | 1.2 | 1.2 | 0.05 |
| Total Epochs | 150 | 150 | 150 | 30 |
| Warm-up Epochs | 60 | 45 | 45 | 0 |
| GPUs used per model | 1 | 8 | 8 | 1 |
| Stopping Accuracy (%) | 100 | 100 | 90 | 100 |

Table 6: Hyperparameters obtained using tuning for CIFAR100

