# OpenReview forum: "Stochastic Weight Averaging in Parallel: Large-Batch Training That Generalizes Well"
_ICLR.cc/2020/Conference — Accept (Poster)_

### Official Review · AnonReviewer1 · 2019-10-18
**Official Blind Review #1**

**Rating:** 6

**Review:**

This paper proposes a parallel version of the stochastic weight averaging method. It utilizes to phases to train the DNN. The first phase consists of distributed large-batch training, where the learning rate is scaled linearly with respect to the scale of the batch size. The second phase consists of using small batches with SWA to obtain the final model. Experiments verify that this method is able to achieve similar generalization performance as small-batch methods in less training time. A comparison against small-batch SWA, large-batch SWA, etc.

Strengths:

The proposed algorithm is a natural extension to SWA, and appears to give good generalization performance. It is able to utilize large-batch training effectively, which is perhaps surprising given the amount of tuning necessary in Shallue, et al. (2018) in order to achieve good performance. The experiments are well-detailed, and some interesting visualizations, graphs, and empirical analyses are provided.

Weaknesses:

I think that this paper is fairly complete; the only question is whether or not it contains enough novelty, as it is a natural extension of SWA to the parallelized setting. No theoretical analysis of the algorithm is given.

Some questions and comments:

- How sensitive is the algorithm to the choice of the transition point? How was the transition point tuned?
- How large of a batch size can one use before this approach breaks down or is no longer efficient?
- In Figure 2, LB is shown to obtain worse training error than SGD. What is the reason for this? This seems contrary to theory (assuming one has already converged to a neighborhood of the solution).
- The authors comment in the conclusion that "it is possible that regions with bad and good generalization performance are connected through regions of low training loss". Can one check if this is related to the invariances described in Dinh, et al. (2017)?
- What is the relationship between SWAP and methods on local SGD?

This paper is missing some classical optimization references on increasing batch sizes, which has been well-studied within that literature:

[1] Byrd, Richard H., et al. "Sample size selection in optimization methods for machine learning." Mathematical programming 134.1 (2012): 127-155.
[2] Bollapragada, Raghu, Richard Byrd, and Jorge Nocedal. "Adaptive sampling strategies for stochastic optimization." SIAM Journal on Optimization 28.4 (2018): 3312-3343.
[3] Bollapragada, Raghu, et al. "A progressive batching L-BFGS method for machine learning." arXiv preprint arXiv:1802.05374 (2018).
[4] Friedlander, Michael P., and Mark Schmidt. "Hybrid deterministic-stochastic methods for data fitting." SIAM Journal on Scientific Computing 34.3 (2012): A1380-A1405.

Although developing some theory for this algorithm would be beneficial, this paper performs a comprehensive set of experiments and is well-written. For these reasons, I'm inclined to accept the paper.

**Experience Assessment:**

I have read many papers in this area.

**Review Assessment: Checking Correctness Of Derivations And Theory:**

I carefully checked the derivations and theory.

**Review Assessment: Checking Correctness Of Experiments:**

I carefully checked the experiments.

**Review Assessment: Thoroughness In Paper Reading:**

I read the paper at least twice and used my best judgement in assessing the paper.

---

> ### Author Response · Authors · 2019-11-14
> **Comment to Review #1**
>
> We thank Reviewer 1 for their thoughtful comments.
>
> Novelty:
>
> We agree that it is a natural extension of SWA to the parallelized setting. However, we believe that this is novel in a few ways:
> It was not obvious to the authors that SWA should improve upon the output found by a large-batch algorithm. We feel that an important contribution is our observation that the initial iterations with large-batch training do not cause permanent damage to the model, and in fact, can initialize small-batch training with SWA for overall more effective training.  This points to the characteristics of the topology of the loss surface that should be better understood.
> Before this work, we did not know if small-batch runs seeded from a large-batch output would provide diverse-enough models for the SWA ensembling to work, and in this work, we have demonstrated that they do.
> Finally, algorithms to accelerate training are of great value to the community. On this front SWAP is an algorithm that efficiently utilizes available resources and is also very simple. We believe that simplicity is a valuable characteristic for such an algorithm.
>
> Other comments:
>
> 1. Choice of transition point: Choosing a transition point between phase 1 and phase 2 presents a trade-off: We would like the large-batch training to cover most of the training process while preventing overfitting. In our experiments, we observed that switching just before the training loss reaches zero, for example, when the training accuracy is 90%, works for the CIFAR-10/100 and ImageNet (Top5 accuracy) datasets. We also noticed that the generalization performance is not very sensitive to the transition point, for instance, in the CIFAR datasets choosing target large-batch training accuracies in the 85% to 98% range yields very similar generalization performance.
>
> 2. How large can the batch-size be?  A thorough experiment to characterize these trade-offs is important but was beyond our computational resources. In our experiments, and for our limited choice of datasets we observe that SWAP has better scaling than simple large-batch training. We expect that in the same manner, when large-batch training shows diminishing returns, SWAP will as well. For example, we observe that, as the large-batch batch-size grows the required number of small-batch epochs in phase 2 increases. Each of these epochs is slower than the large-batch epoch so at some point the time to execute these might dominate.
>
> 3. The point 'LB' in Figure 2 is the output of phase 1. Recall that this point is obtained by early stopping the large-batch training (as discussed above, it can be around 90\% training accuracy). Hence, the training error is worse than SGD. We will modify figure 2 in the revision to clarify this.
>
> 4. We believe that this is not the case. As far as we understand the invariances described in Dinh et al, are changes in the weights that maintain invariant the function that the model computes. Since the function does not change, its behavior in the train and validation sets stays the same. That is to say, if one selects weights invariant in this manner, the generalization behavior of the model would be the same.  We observe something different, the model changes its generalization behavior without having to escape a basin of 'bad generalization' to reach a basin of 'good generalization'.
>
> 5. Connection with local SGD: Thanks for raising this point. This was the primary concern of Reviewer 2. For brevity, we refer the reviewer to our response to reviewer 2, where we provide comprehensive scrutiny on the relation between SWAP and post-local SGD.
>
> As for developing some theory, in Section 4.1, we try to shed some analytical light on our algorithm. Since SWA and SWAP are different ways to sample from the SGD stationary distribution, all the relevant theory contained in the original SWA references can be applied to ours. However, that does not imply that the analysis is complete since these papers use strong simplifying assumptions from Mandt et. al 2017. We believe that improving generalization with weight averaging is a powerful technique, and a complete theoretical analysis is an important open question.

---

### Official Review · AnonReviewer2 · 2019-10-23
**Official Blind Review #2**

**Rating:** 6

**Review:**

This paper proposes a 2-stage SGD variant that improves the generalization. The experiments show good performance.
However, there are some weakness in this paper:

1. (Minor issue) The Update() function in Algorithm 1 seems to be something very general. However, it seems that Update() is simply SGD or SGD with (Nesterov) momentum, according to Section 5.1. Furthermore, the authors never explicitly explain what exact Update() function is used, which is very unfriendly to the readers.

2. The major issue is that the proposed algorithm and the contribution (improvement of generalization) is not novel. Phase 2 of Algorithm 2 is called local SGD, proposed in [1,2]. Local SGD also has variants with Polyak momentum and Nesterov momentum [3]. Furthermore, [4] has already proposed an algorithm, post-local SGD, which is basically the same as Algorithm 1 in this paper (run fully synchronous SGD first and then local SGD). Note that [4] also shows that post-local SGD converges to flatter minima, and results in better generalization. Please correct me if I'm wrong, and explain the difference between Algorithm 1 and (post-)local SGD in details.


----------
Reference

[1] Stich, Sebastian U.. “Local SGD Converges Fast and Communicates Little.” ArXiv abs/1805.09767 (2018).
[2] Yu, Hao et al. “Parallel Restarted SGD with Faster Convergence and Less Communication: Demystifying Why Model Averaging Works for Deep Learning.” AAAI (2018).
[3] Yu, Hao et al. “On the Linear Speedup Analysis of Communication Efficient Momentum SGD for Distributed Non-Convex Optimization.” ICML (2019).
[4] Lin, Tao et al. “Don't Use Large Mini-Batches, Use Local SGD.” ArXiv abs/1808.07217 (2018).


**Experience Assessment:**

I have read many papers in this area.

**Review Assessment: Checking Correctness Of Derivations And Theory:**

N/A

**Review Assessment: Checking Correctness Of Experiments:**

I assessed the sensibility of the experiments.

**Review Assessment: Thoroughness In Paper Reading:**

I read the paper at least twice and used my best judgement in assessing the paper.

---

> ### Author Response · Authors · 2019-11-11
> **Comment to review #2**
>
> We believe SWAP and post-local SGD are in fact distinct: in the mechanisms that must power them, in the conclusions that we can draw about the dynamics of DNN optimization from them; and furthermore, can be used complementarily.
> The practical distinction is that post-local SGD synchronizes every tens of iterations. For example, authors in [4] average after 32 iterations for CIFAR10 (Figure 4(a)) and after 8 iterations in ImageNet (Figure 9 in Appendix B.3.2). On the other hand, in SWAP experiments we synchronize "only once" after thousands of iterations (2910 iterations in CIFAR10 and 20,794 iterations on imagenet).
> This difference of 3 orders of magnitude likely indicates that different mechanisms are behind the success of SWAP and the success of post-local SGD. We believe that the phenomena we observe in SWAP is unlikely to be explained by the arguments in [1,2,3,4] where the divergence between local models is assumed to be controlled and thus the averaged model results from noisy gradient updates with gradient noise of similar shape but larger magnitude [4].
> While it is believable that Post-local SGD improves on generalization by a mechanism that noisifies the gradient  with the correct noise structure (thus mimicking the behavior of small-batch SGD). SWAP seems to be taking advantage of different mechanics.  For example, a mechanism that ensembles models sampled independently from an approximate posterior distribution over weights seems more likely.
> In any case, we believe the community would benefit from having access to both these observations, for they both are likely to hint at underlying mechanics that need to be better understood. We believe that the fact that SWAP works at all is surprising and further evidence that the strategies of Stochastic Weight Averaging are robust and valuable.
>
> We agree with the reviewer that not including references to local-SGD in the prior work section and a clear comparison of the work is a bad oversight, and mean to address it.
>
> On the minor comment 1:
> We agree that our choice makes that section unfriendly to the reader.  We will modify the manuscript to specify that here Update() is in fact simply SGD (with momentum and weight decay).
> This is a mistake that stemmed from the fact that we observed similar behavior when other first-order stochastic methods are used, for example, we observed commensurate speedups with Adam.
> Since the best behaving version was simply SGD with momentum and weight decay, we didn’t include these in the manuscript. We will write a more precise version of our pseudocode.

---

### Official Review · AnonReviewer3 · 2019-10-23
**Official Blind Review #3**

**Rating:** 3

**Review:**

In the paper, the authors propose a novel three-stage training strategy for deep learning models, training using large batch, training using small-batch locally and then aggregate models. Experimental results show that the proposed method converges faster than compared methods.  I have the following concerns:

1) In figure 1, It looks like that the local models in worker 1, 2, 3 can reach the same testing accuracy without average. What is the meaning of computing average in this case?

2) In the paper, authors mentioned that “Note that there exist training schemes in the literature that train on even larger batch sizes such as 32k (You et al., 2017; Jia et al., 2018), but these methods require a lot of hyperparameter tuning specific to the dataset.” As far as I know, they just need to tune warmup steps and peak learning rate, which is also required in the paper. In the proposed method, it is required to tune the switch point between phase 1 and phase 2.

3) In the experiment, it also uses warmup for small batch size, is it necessary?

4) Does the large batch training use layer-wise learning rate scaling? From my point of view, it is better to use it in the large-batch training.

5) A guideline about how to select the switch point between phase 1 and phase 2 should be given if it takes time to tune it.


**Experience Assessment:**

I have published one or two papers in this area.

**Review Assessment: Checking Correctness Of Derivations And Theory:**

I assessed the sensibility of the derivations and theory.

**Review Assessment: Checking Correctness Of Experiments:**

I assessed the sensibility of the experiments.

**Review Assessment: Thoroughness In Paper Reading:**

I read the paper at least twice and used my best judgement in assessing the paper.

---

> ### Author Response · Authors · 2019-11-14
> **Comments to Review #3**
>
> Thank you for the feedback on your concerns. Here are our clarifications:
>
> 1. (Figure 1): The difference in final test accuracies between the independent and averaged models is around 0.53% (Table 1). This is barely noticeable in Figure 1 where the y-axis goes from 10% to 100%. For CIFAR100, this difference, is around ~2.5% (Table 2). In our figure we intended to show how a hypothetical averaged model behaves better than each worker’s model. But agree we could have also shown a version where the final accuracy of the averaged model is obviously higher than the other ones.
>
> 2. (Existing large-batch training results): The blanket statement that such methods require a "lot" of hyperparameter tuning is a mistake, that phrase will be removed. (You et al., 2017)  introduces only one new hyperparameter (scaling factor l).
>
> 3. (Warm-up for small-batch sizes):  We concluded that using warmup for small-batch SGD was advantageous for it. We chose the hyperparameters via a grid search over warmup epochs and peak learning rate. The best settings were achieved when the peak learning rate is larger than the initial learning rate (See appendix A).
>
> 4. (LARS with SWAP): Our intention is twofold, to provide an algorithm and to highlight a phenomenon. Since it is not clear why large-batch SGD should worsen the test behavior, and it is in itself an important question, we rather design experiments that probe this phenomenon. If we were to use LARS or any other choice we’ll make the interpretation harder. That is not to say that the best possible algorithm would use different strategies for a large and small batch (for example LARS, mixed-precision + LARS, local-SGD NovoGrad, etc) and we intend to explore this in a sequel.
>
> 5. (Switching point between phases 1 and 2): In our experience, SWAP lent itself to a simple tuning strategy which we are happy to describe: We choose a threshold training accuracy a few percentage points lower than saturation (in our experiments 90% was effective) and switch phases when that accuracy is achieved. We agree that sharing our experience might be useful for someone who might try to apply the method.

---

### Public Comment · ~Scarlett_Li1 · 2019-10-24
**Why only compare to SGD?**

SDG is not a strong baseline, why only compare to SGD? Thank you.

---

> ### Public Comment · ~Sairaam_Venkatraman1 · 2019-10-24
> **SGD isn't really that weak**
>
> Hi!
> First, I am not an author - for clarity on all sides. I wouldn't consider SGD to be a weak baseline by any means, as it is still in current use. Further, there is experimental proof that SGD does better than optimisers such as Adam or RMSprop in some models [1]. I assume that you mean SGD is "weaker", when compared to other optimisers.
>
> [1] https://arxiv.org/abs/1712.07628

---

### Decision · Program_Chairs · 2019-12-19

**Decision:**

Accept (Poster)

**Comment:**

The authors proposed a simple and effective approach to parallel training based on stochastic weight averaging. Moreover, the authors have carefully addressed the reviewer comments in the discussion period, particularly the relation to local SGD, to the satisfaction of reviewers. Local SGD mimics sequential SGD with noise induced by lack of synchronization, whereas SWAP averages multiple samples from a stationary distribution, and synchronizes at the end. Please clarify these points and carefully account for reviewer comments in the final version. Overall, the proposed approach will make an excellent addition to the program, both elegant and practically useful.